# Seismological Problem, Seismic Waves and the Seismic Mainshock

**Bogdan Felix Apostol** 

Department of Engineering Seismology, Institute for Earth's Physics, Magurele-Bucharest MG-6,
P.O. Box MG-35, 077125 Magurele, Romania; afelix@theory.nipne.ro

**Abstract:** The elastic wave equation with seismic tensorial force is solved in a homogeneous and isotropic medium (the Earth). Spherical-shell waves are obtained, which are associated to the primary *P* and *S* seismic waves. It is shown that these waves produce secondary waves with sources on the plane surface of a half-space, which have the form of abrupt walls with a long tail, propagating in the interior and on the surface of the half-space. These secondary waves are associated to the seismic mainshock. The results, previously reported, are re-derived using Fourier transformations and specific regularization procedures. The relevance of this seismic motion for the ground motion, the seismographs' recordings and the effect of the inhomogeneities in the medium are discussed.

**Keywords:** primary seismic waves; seismic mainshock; elastic wave equation; seismic tensorial force; regularization

**MSC:** 35C05; 35C07; 35D99

## 1. Introduction

A typical seismogram recorded on the Earth's surface consists of a faible tremor followed by an abrupt motion with a long tail [1]. The precursory tremor is associated with spherical-shell waves, called primary seismic waves, while the abrupt motion is known as the seismic mainshock. Such a seismogram is sketched in Figure 1. The interpretation of seismograms has been recognized since long as the Seismological Problem (or Lamb's problem) [2]. It is known that this seismic motion originates in a very small focal region, where a short, sudden disturbance occurs. In the absence of knowledge of the force acting in the seismic focus, the primary waves are derived via the so-called double-couple procedure, based on the solution to the Stokes problem [3]. Apart from inconvenient restrictions to particular orientations of the double couple, the result may include unphysical contributions [4,5]. The mainshock, associated with Rayleigh surface waves [6], is treated as a vibration problem [7,8].

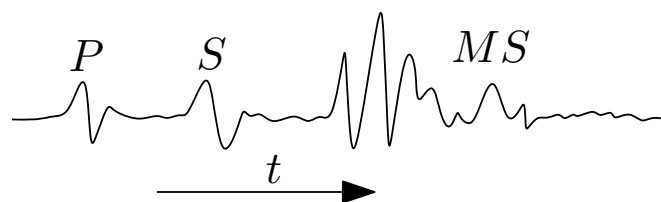

**Figure 1.** Schematic representation of a typical seismogram, with the *P* and *S* waves and the mainshocks *MS*; the arrow indicates the flow of the time *t*.

The force density acting in the seismic focus has been introduced in Ref. [9]. It reads

$$f_i = T M_{ij} \delta(t) \partial_j \delta(\boldsymbol{R} - \boldsymbol{R}_0) \ , \tag{1}$$

where $M_{ij}$ are the Cartesian components of a symmetrical tensor, known as the tensor of the seismic moment; this force acts in a very short time *T* in the seismic focus localized

at position $\boldsymbol{R}_0$. It corresponds to a shearing fault. The total force and the total angular momentum of this force density are zero, according to the physical requirements. The primary $P$ (longitudinal) and $S$ (transverse) seismic waves produced by this force in a homogeneous and isotropic elastic body have been derived, in agreement with the recorded seismograms [9]. The force given by Equation (1) corresponds to a single rupture in the focus; several successive ruptures may appear (for a so-called structured focus), with corresponding oscillations displayed by the seismic waves.

For limited distances, the Earth may be approximated by a half-space with a plane surface. Once arrived at the Earth's surface, the primary waves generate surface sources which, in turn, produce secondary waves, according to Huygens' principle. These secondary waves have the form of an abrupt wall with a long tail (actually two walls, corresponding to the two primary waves), in agreement with the mainshock exhibited by seismograms [9]. A structured focus may generate oscillations in the mainshock.

Also, the static deformations of a (homogeneous and isotropic) elastic half-space generated by the tensorial force density given by Equation (1) have been computed, as well as the vibrations of the half-space [10]. Moreover, the seismic moment tensor was derived from measurements of the primary waves at Earth's surface (the so-called Inverse Seismological Problem) [10].

The Seismological Problem is an old problem. On one hand, its solution should respond to our need to know the form of the seismic waves that appear on the Earth's surface during an earthquake, and, on the other, we hope to know the characteristics of the focal region from measurements of the seismic waves. Basic answers to this problem are included in classical books (see, for instance Refs. [4,5]). Great progress has been made in recent times in analyzing methods of solution, as, for example, in Refs. [11–14], where important new insights have been obtained on Lamb's problem for an elastic half-space. The method of solution consists in using Green functions for the elastic wave equation with combinations of Stokes solutions for double-couple elastic forces. The results depend on the particular orientation of the couples. A fully covariant solution would require the use of a tensorial force like that given by Equation (1), which was introduced only recently in Refs. [9,10]. In general, any method of solving the equation of elastic waves for a localized focus implies unphysical contributions, which should be regularized. The reason for such a rather special situation is the singular nature of the seismic source. In many cases, the standard solution includes an undesired continuous displacement between the wavefronts of the two primary waves, which should be removed. The regularization is difficult to apply on particular form of solution. In Ref. [9] a scheme of regularization is used for the decomposition of the solution in Helmholtz potentials and the use of the Kirchhoff formula. In the present paper, we use another method, based on Fourier transformations, which, besides the standard regularization, requires an additional Coulomb-type regularization. By using this new type of regularization, we re-obtain the previous results, which shows, on one hand, the correctness of the results, and, on the other, the complexity of the problem. The primary waves given by Equations (27) and (28) are new. Their scissor-like shape for large distances are in perfect agreement with the recorded seismograms.

Moreover, the primary waves generate sources of (secondary) elastic waves on Earth's surface, which give rise to the mainshock, as observed in all seismograms. The standard approach to this problem is the analysis of the surface Rayleigh waves, which leads to the identification of many useful resonances. However, this is the typical method for a vibration problem, while we would like to obtain the wall-like structure of a propagating mainshock, which is a wave. The path to solving this problem is opened by the mainshock equation discussed in this paper (Equation (29)), which has been introduced recently in Ref. [9]. In that reference an approximate method of solution is used for the wave equation and the corresponding Kirchhoff formula. In the present paper it is shown that by using Fourier transformations, the problem can be reduced to a Weyl–Sommerfeld integral, which leads to the same results. The wall-like structure of the mainshock is described by Equation (47) and sketched in Figure 2.

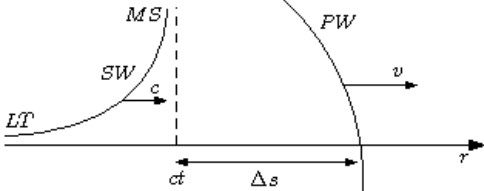

**Figure 2.** Primary wave (PW) moving with velocity $v$ on the Earth's surface, and secondary wave (SW) moving with velocity $c < v$, the mainshock (MS) and the long tail (LT, Equation (47) for $z = 0$); the separation between the two wavefronts is $\Delta s = 2(v - c)t$ and the time delay is $\Delta t = (2r/c)(v/c - 1)$, where $r$ is the epicentral distance.

All the results presented in this paper are new. The advantage of the methods described here consists of a more exact formulation of the Seismological Problem, and a consistent use of standard mathematical procedures. All these reveal the complexity of the problem and the richness of the properties of its solution. The paper ends with a brief presentation of the problem of the inhomogeneities in an elastic medium, which is of great importance in seismology.

## 2. Elastic Wave Equation

Seismic waves are governed by the Navier–Cauchy equation [15]

$$\ddot{u}_i - c_t^2 \partial_j \partial_j u_i - (c_l^2 - c_t^2) \partial_i \partial_j u_j = T m_{ij} \delta(t) \partial_j \delta(\mathbf{R}) \ , \tag{2}$$

where $u_i$ ($u_j$, $i, j = 1, 2, 3$) are the Cartesian components of the local displacement vector, $c_{t,l}$ are the transverse and the longitudinal elastic wave velocities (in a homogeneous and isotropic elastic body), $m_{ij}$ is the symmetrical tensor of the seismic moment divided by the density of the medium and $T$ denotes the short duration of the force localized at the initial moment $t = 0$ in the seismic focus placed at $\mathbf{R} = 0$ [9].

Equation (2) was solved for the seismic waves in Ref. [9] by using the decomposition in Helmholtz potentials and the Kirchhoff formula. A certain regularization procedure was needed in order to remove unphysical contributions arising from the singular nature of the source term in Equation (2). We describe here a different method of solving the above equation, which throws more light upon the singular, unphysical behavior of the solution.

A direct way to solve Equation (2) is to perform a Fourier transform, which gives

$$(\omega^2 - c_t^2 k^2) u_i - (c_l^2 - c_t^2) k_i k_j u_j = -i T m_{ij} k_j \ , \tag{3}$$

where $k_i$ ($k_j$) are the Cartesian components of the wavevector $\mathbf{k}$; the arguments $\omega$ and $\mathbf{k}$ of the Fourier transforms in Equation (3) are omitted for brevity. From this equation, we obtain

$$k_i u_i = -\frac{i T m_{ij} k_i k_j}{\omega^2 - c_l^2 k^2} \ , \tag{4}$$

which, inserted in Equation (3), gives

$$u_i = -\frac{i T m_{ij} k_j}{\omega^2 - c_t^2 k^2} - (c_l^2 - c_t^2) \frac{i T m_{jk} k_i k_j k_k}{(\omega^2 - c_t^2 k^2)(\omega^2 - c_l^2 k^2)} \ . \tag{5}$$

We denote the fractions on the right in this equation by $u_i^{(1,2)}$ and perform the reverse Fourier transform. We obtain

$$u_i^{(1)} = -\frac{iT}{(2\pi)^4} \int d\omega d\mathbf{k} \frac{m_{ij}k_j}{\omega^2 - c_t^2 k^2} e^{-i\omega t} e^{i\mathbf{k}\mathbf{R}} =$$

$$= -\frac{T}{(2\pi)^4} m_{ij}\partial_j \int d\omega d\mathbf{k} \frac{e^{-i\omega t} e^{i\mathbf{k}\mathbf{R}}}{(\omega - c_t k + i\varepsilon)(\omega + c_t k + i\varepsilon)} = \tag{6}$$

$$= \frac{T}{4\pi c_t} m_{ij}\partial_j \frac{\delta(R - c_t t)}{R} \ ,$$

where we placed the $\omega$-poles in the lower half-plane ($\varepsilon \to 0^+$), in order to obtain waves which obey the causality principle (i.e., they are vanishing for $t < 0$).

A similar procedure for $u_i^{(2)}$ gives

$$u_i^{(2)} = \frac{T}{2c_t(2\pi)^2} m_{jk}\partial_i\partial_j\partial_k \frac{1}{R} \int_{-\infty}^{+\infty} dk \frac{1}{k^2} e^{ik(R - c_t t)} - (t \to l) \ . \tag{7}$$

We can see that the $k$-integral in Equation (7) is improper. We need to give a meaning to this integral.

We may use several procedures to regularize this integral. For instance, we note that its second derivative is a Dirac $\delta$-function, so we may integrate the $\delta$-function twice, with two constants which need to be determined. Another procedure might be using the integral

$$\int_{-\infty}^{+\infty} dk \frac{1}{k} e^{ik(R - c_t t)} \tag{8}$$

as a principal value; it is $i\pi sgn(R - c_t t)$, and we can integrate it with a constant to be determined. Also, we may view the integral in Equation (8) as giving a stepwise $\theta$-function. All these regularization procedures give different results, and they need a justification.

### 3. Coulomb Potential Regularization

Let us introduce the function

$$F = -\frac{T}{2c(2\pi)^2} \frac{1}{R} \int_{-\infty}^{+\infty} dk \frac{1}{k^2} e^{ik(R - ct)} \ . \tag{9}$$

The solution $u_i^{(2)}$ reads

$$u_i^{(2)} = m_{jk}\partial_i\partial_j\partial_k(F_l - F_t) \ , \tag{10}$$

where $F_{l,t}$ is obtained by replacing $c$ with $c_{l,t}$. The factor $1/k^2$ in Equation (9) may be viewed as the Fourier transform of the Coulomb potential. Indeed,

$$\int d\mathbf{R} \frac{1}{R} e^{-i\mathbf{k}\mathbf{R}} e^{-\mu R} = \frac{4\pi}{k^2 + \mu^2} \ , \tag{11}$$

where $\mu \to 0^+$ is a small cutoff. This may suggest that $F$ should be regularized by

$$F = -\frac{T}{2c(2\pi)^2} \frac{1}{R} \int_{-\infty}^{+\infty} dk \frac{1}{k^2 + \mu^2} e^{ik(R - ct)} \ . \tag{12}$$

On the other hand, using direct calculations we obtain from Equation (9)

$$\ddot{F} = c^2 \Delta F = \frac{cT}{4\pi} \frac{1}{R} \delta(R - ct) \ ; \tag{13}$$

in the limit $R \rightarrow 0$, the term $c^2 \Delta F$ dominates, in comparison with $\ddot{F}$, so $F$ satisfies the equation

$$\ddot{F} - c^2 \Delta F = -\frac{cT}{4\pi} \frac{1}{R} \delta(ct) = -\frac{T}{4\pi} \frac{1}{R} \delta(t) \ . \tag{14}$$

If we integrate this equation with respect to time, we obtain

$$\Delta \int dt F = \frac{T}{4\pi c^2} \frac{1}{R} \ ; \tag{15}$$

indeed, from Equation (9) we have

$$\int dt F = -\frac{T}{4\pi c^2} \frac{1}{R} \int dk \frac{\delta(k)}{k^2 + \mu^2} e^{ikR} \tag{16}$$

and

$$\Delta \int dt F = \frac{T}{4\pi c^2} \frac{1}{R} \int dk \frac{k^2 \delta(k)}{k^2 + \mu^2} e^{ikR} =$$

$$= \frac{T}{4\pi c^2} \frac{1}{R} \int dk \delta(k) e^{ikR} = \frac{T}{4\pi c^2} \frac{1}{R} \ . \tag{17}$$

It is easy to see that the $\mu$-regularization does not work for the function $F$ given by Equation (12), because we have already imposed the retarded wave condition, while the $\mu$-regularization requires the presence of both retarded and advanced waves (according to the regularization of the static Coulomb potential, Equation (11)). Consequently, we must derive the function $F$ from its wave Equation (14), and retain only the retarded solutions. By Fourier transforming Equation (14), we obtain

$$F = \frac{T}{(k^2 + \mu^2)(\omega^2 - c^2 k^2)} \tag{18}$$

and

$$F = \frac{T}{(2\pi)^4} \int d\omega d\mathbf{k} \frac{e^{-i\omega t} e^{i\mathbf{k}\mathbf{R}}}{(k^2 + \mu^2)(\omega^2 - c^2 k^2)} =$$

$$= -\frac{T}{2c(2\pi)^2 R} \int \frac{dk}{k^2 + \mu^2} \left[ e^{ik(R-ct)} - e^{ik(R+ct)} \right] \ , \tag{19}$$

where we placed the $\omega$-poles in the lower half-plane, according to the causality principle (such that $F = 0$ for $t < 0$). We can see that the term $e^{ik(R+ct)}$ gives, in fact, a damped contribution $e^{-\mu(R+ct)}$, although, formally, it looks like an advanced wave. The result of the integration in Equation (19) is

$$F = -\frac{T}{8\pi c R \mu} \left\{ \begin{array}{l} e^{\mu(R-ct)} - e^{-\mu(R+ct)} \ , \ 0 < R < ct \ , \\ e^{-\mu(R-ct)} - e^{-\mu(R+ct)} \ , \ 0 < ct < R \ . \end{array} \right\} \ . \tag{20}$$

Here, we may take the limit $\mu \rightarrow 0$ and obtain

$$F = -\frac{T}{4\pi c} \left[ \theta(ct - R) + \frac{ct}{R} \theta(R - ct) \right] \ , \tag{21}$$

for the retarded wave, where $\theta(0) = 1/2$; the value $1/2$ is expected for a series of continuous functions which approximate the stepwise $\theta$-function. This result was previously obtained in Ref. [9], by solving Equation (14) with the Kirchhoff retarded potentials (where the function $F$ was introduced and Equation (14) established using the Helmholtz potentials for the Navier–Cauchy equation).

## 4. Seismic Wave Regularization

First, we note that the function $F$ given by Equation (21) for $ct \neq R$ satisfies the free-wave equation. Therefore, the $\theta$-contributions should be removed, and only the

contributions for $ct = R$ should be retained. This is valid also for the derivatives of the prefactor of the $\theta$-function. The first-order spatial derivative is

$$\partial_i F = \frac{T}{8\pi c} \left[ \frac{x_i}{R^2}(R - ct)\delta(R - ct) + \frac{2ctx_i}{R^3}\theta(R - ct) \right] , \qquad (22)$$

where we introduced the factor $1/2$ for $\theta(0) = 0$. The first term in the above equation is zero, while its derivatives are not; the second term should be disregarded, except for $R = ct$. The second-order derivative is

$$\partial_i \partial_j F = \frac{T}{8\pi c} \left( \frac{\delta_{ij}}{R} - \frac{ct\delta_{ij}}{R^2} - \frac{x_i x_j}{R^3} + \frac{3ctx_i x_j}{R^4} \right)\delta(R - ct) +$$

$$+ \frac{T}{8\pi c} \left( \frac{x_i x_j}{R^2} - \frac{ctx_i x_j}{R^3} \right)\delta'(R - ct) + \qquad (23)$$

$$+ \frac{T}{4\pi c} \left( \frac{ct\delta_{ij}}{R^3} - \frac{3ctx_i x_j}{R^5} \right)\theta(R - ct) .$$

From Equation (23), we obtain

$$\Delta F = \frac{T}{4\pi c} \frac{\delta(R - ct)}{R} + \frac{T}{8\pi c} \left( 1 - \frac{ct}{R} \right)\delta'(R - ct) , \qquad (24)$$

which differs from Equation (13) by the $\delta'$-contribution. Consequently, the $\delta'$-contribution must be removed from $\partial_i \partial_j F$. Also, according to the discussion above, the $\theta$-contribution must be removed, so we are left with the regularized expression

$$\partial_i \partial_j F = \frac{T}{8\pi c} \left( \frac{\delta_{ij}}{R} - \frac{ct\delta_{ij}}{R^2} - \frac{x_i x_j}{R^3} + \frac{3ctx_i x_j}{R^4} \right)\delta(R - ct) . \qquad (25)$$

The regularization procedure described above amounts to viewing the function $\delta(R - ct)$ as a function peaked on $R = ct$, of the order $1/l$ over a small distance $l$, and zero otherwise. Similarly, the function $\delta'(R - ct)$ is of the order $1/l^2$ extended over $l$. Indeed, this way, the $\delta'$-function in Equation (23) brings a small contribution, which may be neglected.

Now, it is easy to compute $m_{jk}\partial_i \partial_j \partial_k$ in Equation (10). In general, $m_{jk}\partial_k$ may be replaced by an external force $f_j$, which is applied to $\partial_i \partial_j F$. It is not permissible to set $ct = R$ in Equation (25), because $f_j$ may include derivatives, which, for the prefactor in Equation (25), should be computed before setting $ct = R$. For the derivative of the $\delta$-function, we should put $ct = R$ in its prefactor, in accordance with the regularization of the quantity $\partial_i \partial_j F$ discussed above. We obtain

$$m_{jk}\partial_i \partial_j \partial_k =$$

$$= \frac{T}{8\pi cR^3} \left( m_{jj}x_i + 4m_{ij}x_j - \frac{9m_{jk}x_i x_j x_k}{R^2} \right)\delta(R - ct) + \qquad (26)$$

$$+ \frac{T}{4\pi c} \frac{m_{jk}x_i x_j x_k}{R^4}\delta'(R - ct) ,$$

where we set $ct = R$ in the $\delta'$-contribution.

By making use of Equations (6) and (10), we obtain the near-field displacement

$$u_i^n = -\frac{T m_{ij} x_j}{4\pi c_t R^3} \delta(R - c_t t) +$$

$$+ \frac{T}{8\pi R^3}\left( m_{jj} x_i + 4 m_{ij} x_j - \frac{9 m_{jk} x_i x_j x_k}{R^2} \right) \cdot$$

$$\cdot\left[ \frac{1}{c_l}\delta(R - c_l t) - \frac{1}{c_t}\delta(R - c_t t) \right] \tag{27}$$

and the far-field displacement ($R \gg l$)

$$u_i^f = \frac{T m_{ij} x_j}{4\pi c_t R^2} \delta'(R - c_t t) +$$

$$+ \frac{T m_{jk} x_i x_j x_k}{4\pi R^4}\left[ \frac{1}{c_l}\delta'(R - c_l t) - \frac{1}{c_t}\delta'(R - c_t t) \right] . \tag{28}$$

We can see that the far-field displacement consists of two spherical-shell waves with a scissor-like shape, one longitudinal, propagating with velocity $c_l$, the other transverse, propagating with velocity $c_t$. These are the $P$ and $S$ seismic waves [9]. The relevance of the near-field displacement for the derivation of the seismic moment tensor has been discussed in Ref. [10].

## 5. Secondary Waves

The focus of a typical earthquake is localized both in space and time, in a point beneath the Earth's surface. During the short time of releasing the seismic energy in an earthquake, the focus produces two primary waves, which look like spherical shells, propagating with the longitudinal and transverse elastic wave velocities $c_{l,t}$. These primary waves are known in seismology as the $P$ and $S$ seismic waves. Once arrived at the Earth's surface, such a primary wave generates a circular wavefront on the surface, which propagates with a velocity $v$, greater than the velocity $c$ of the primary wave. Indeed, it is easy to see, from their definitions, that $v = cR/r$, where $R$ is the Earth's radius and $r$ is the epicentral distance. The difference between the two velocities goes to zero for large epicentral distances. These wavefronts are localized on the surface in an infinitesimal torus. According to the Huygens principle, they generate secondary waves, which give the seismic mainshock. The displacement is given by the derivatives of some potentials, denoted here generically by $\psi$. These potentials satisfy the wave equation

$$\ddot{\psi} - c^2 \Delta \psi = \delta(r - vt)\delta(z) , \tag{29}$$

where $r$ is the position vector parallel to the surface and $t$ denotes the time; the surface is viewed as a plane surface, placed at $z = 0$. The velocity $v$ is considered constant. This equation can be called the mainshock equation. The (homogeneous and isotropic) elastic medium occupies the half-space $z < 0$. The above equation is valid for a limited range of epicentral distances, $r$, centered on a value of the order of the depth of the focus [9].

We introduce the notation

$$F(\boldsymbol{R}, t) = \delta(r - vt)\delta(z) , \tag{30}$$

where $\boldsymbol{R} = (r, z)$, and compare Equation (29) with the same equation with the source $S = \delta(\boldsymbol{R})\delta(t) = \delta(\boldsymbol{r})\delta(z)\delta(t)$. The solution of this latter equation is the spherical wave $\delta(R - ct)/4\pi cR$. We note that source $S$ is singular in a point with four coordinates (time included), while source $F$ is singular in a set of points, each with three coordinates, placed along a line ($r = vt$). Therefore, source $F$ is more singular than source $S$, so we expect a divergent solution of Equation (29). The singularity is more effective for a larger size of the length of the line $r = ct$, i.e., for large $r$, so we need, at least, a small cutoff wavevector.

We show below that Equation (29) has a regular solution for $v \simeq c$, when the source may be treated as a boundary condition, in accordance with the standard procedure.

We perform Fourier transform on Equation (29),

$$(\omega^2 - c^2 K^2)\psi(\omega, \mathbf{K}) = -F(\omega, \mathbf{K}) \ , \tag{31}$$

where $\mathbf{K} = (\mathbf{k}, \kappa)$ and

$$F(\omega, \mathbf{K}) = \int dt d\mathbf{R} \delta(r - vt)\delta(z)e^{i\omega t}e^{-i\mathbf{K}\mathbf{R}} =$$

$$= \frac{1}{v} \int d\mathbf{r}e^{i\omega r/v}e^{-i\mathbf{k}\mathbf{r}} = \frac{2\pi}{v} \int dr \cdot r e^{i\omega r/v} J_0(kr) \ , \tag{32}$$

where $J_0(kr)$ is the Bessel function. This expression can also be written as

$$F(\omega, \mathbf{K}) = \frac{2\pi}{v} \frac{\partial}{\partial(i\omega/v)} \int d r e^{i\omega r/v} J_0(kr) =$$

$$= -\frac{2\pi i}{vk^2} \frac{\partial}{\partial \lambda} \int_0^\infty dx e^{i\lambda x} J_0(x) \ , \tag{33}$$

where $\lambda = \omega/vk$. The integral in the second row of Equation (33) is the Weyl–Sommerfeld integral,

$$I(\lambda) = \int_0^\infty dx e^{i\lambda x} J_0(x) = \frac{\theta(1 - |\lambda|)}{\sqrt{1 - \lambda^2}} + i \text{sgn}(\lambda)\frac{\theta(|\lambda| - 1)}{\sqrt{\lambda^2 - 1}} \ . \tag{34}$$

Therefore, from Equation (31), we obtain

$$\psi(\mathbf{R}, t) = \frac{i}{(2\pi)^3 v} \int d\omega d\mathbf{K} \frac{\partial I/\partial \lambda}{k^2(\omega - cK + i\varepsilon)(\omega + cK + i\varepsilon)} e^{-i\omega t} e^{i\mathbf{K}\mathbf{R}} \ , \tag{35}$$

where $\varepsilon \to 0^+$. We place the $\omega$-poles in the lower half-plane in order to have $\psi = 0$ for $t < 0$, according to the causality principle. Henceforth, we consider $t > 0$ only. Equation (35) can also be written as

$$\psi(\mathbf{R}, t) = \frac{i}{(2\pi)^3 v^2} \int d\lambda d\mathbf{K} \frac{\partial I/\partial \lambda}{k^3(\lambda - \lambda_1)(\lambda - \lambda_2)} e^{-ivkt\lambda} e^{i\mathbf{K}\mathbf{R}} \ , \tag{36}$$

where $\lambda_1 = cK/vk - i\varepsilon$ and $\lambda_2 = -cK/vk - i\varepsilon$. We need to compute $\partial I/\partial \lambda$ for $\lambda_{1,2}$ using Equation (34). We obtain

$$(\partial I/\partial \lambda)_{\lambda_1} = \begin{cases} \frac{\lambda_0}{(1 - \lambda_0^2)^{3/2}} \ , & \lambda_0 = cK/vk < 1 \ , \\ -\frac{i\lambda_0}{(\lambda_0^2 - 1)^{3/2}} \ , & \lambda_0 = cK/vk > 1 \end{cases} \tag{37}$$

and

$$(\partial I/\partial \lambda)_{\lambda_2} = \begin{cases} -\frac{\lambda_0}{(1 - \lambda_0^2)^{3/2}} \ , & \lambda_0 = cK/vk < 1 \ , \\ -\frac{i\lambda_0}{(\lambda_0^2 - 1)^{3/2}} \ , & \lambda_0 = cK/vk > 1 \ . \end{cases} \tag{38}$$

It follows that

$$\psi(\mathbf{R}, t) = \frac{1}{(2\pi)^2 v^2} \int d\mathbf{K} \frac{1}{k^3} \left\{ \begin{array}{c} \frac{\cos cKt}{(1 - \lambda_0^2)^{3/2}} \\ -\frac{\sin cKt}{(\lambda_0^2 - 1)^{3/2}} \end{array} \right\} e^{i\mathbf{K}\mathbf{R}} =$$

$$= \frac{v}{(2\pi)^2} \int d\mathbf{K} \left\{ \begin{array}{c} \frac{\cos cKt}{(v^2 k^2 - c^2 K^2)^{3/2}} \\ -\frac{\sin cKt}{(c^2 K^2 - v^2 k^2)^{3/2}} \end{array} \right\} e^{i\mathbf{K}\mathbf{R}} \ , \tag{39}$$

where the expressions under the square root are positive.

The function $\psi(\boldsymbol{R}, t)$ given by Equation (39) is a superposition of plane waves $e^{icKt+i\boldsymbol{KR}}$ and $e^{-icKt+i\boldsymbol{KR}}$; we must retain only the outgoing wave $e^{-icKt+i\boldsymbol{KR}}$, in accordance with the causality principle, so Equation (39) becomes

$$\psi(\boldsymbol{R}, t) = \frac{v}{2(2\pi)^2} \int d\boldsymbol{K} \left\{ \begin{array}{c} \frac{1}{(v^2 k^2 - c^2 K^2)^{3/2}} \\ -\frac{i}{(c^2 K^2 - v^2 k^2)^{3/2}} \end{array} \right\} e^{-icKt} e^{i\boldsymbol{KR}} \tag{40}$$

(where in the second row we must add the complex conjugate and divide by 2). Also, the wave should be progressive, i.e., $ct > R$, a condition which can also be written as

$$c^2 t^2 > R^2 = r^2 + z^2 \; ; \tag{41}$$

otherwise, the wave is zero.

### 6. Mainshock

For $v$ close to $c$, only the second row in Equation (40) is valid. By using $\gamma^2 = v^2/c^2 - 1$, Equation (40) can be written as

$$\psi(\boldsymbol{R}, t) = -\frac{iv}{2(2\pi)^2 c^3} \int dk d\kappa \frac{e^{-ic\sqrt{\kappa^2 + k^2} t}}{(\kappa^2 - \gamma^2 k^2)^{3/2}} e^{i\kappa z} e^{ikr} =$$

$$= -\frac{iv}{4\pi c^3} \int dk k J_0(kr) \int d\kappa \frac{1}{(\kappa^2 - \gamma^2 k^2)^{3/2}} e^{-ic\sqrt{\kappa^2 + k^2} t} e^{i\kappa z} \; . \tag{42}$$

For $z < 0$ the $\kappa$-integration must be carried out in the lower half-plane. The integrand has two branch points at $\kappa = \pm\gamma k$. It is easy to see that the integral along this branch cut is singular, as expected. According to the discussion above, we put $v = c$ and displace the pole $\kappa = 0$ slightly below in the lower half-plane. This operation provides the standard procedure of treating the source as a boundary condition. For $v = c$ ($\gamma = 0$), the integral in Equation (42) becomes

$$\psi(\boldsymbol{R}, t) \simeq -\frac{i}{4\pi c^2} \int dk k J_0(kr) \int d\kappa \frac{1}{\kappa^3} e^{-ic\sqrt{\kappa^2 + k^2} t} e^{i\kappa z} \; . \tag{43}$$

The pole placed slightly below $\kappa = 0$ plays the role of a lower cutoff wavevector. The calculation of this contribution is performed by writing

$$e^{-ic\sqrt{\kappa^2 + k^2} t} e^{i\kappa z} = e^{-ictk} e^{-ict\kappa^2/2k} e^{i\kappa z} \simeq$$

$$\simeq e^{-ictk} \left( 1 + i\kappa z - \frac{\kappa^2 z^2}{2} - i\frac{ct\kappa^2}{2k} + \dots \right) , \tag{44}$$

which leads to

$$\psi(\boldsymbol{R}, t) \simeq \frac{i}{8\pi c^2} \{ z^2 \int dk k J_0(kr) e^{-ictk} +$$

$$+ ict \int dk J_0(kr) e^{-ictk} \} \int d\kappa \frac{1}{\kappa} \; . \tag{45}$$

Straightforward calculations give

$$\psi(\boldsymbol{R}, t) \simeq \frac{i}{4c^2} \{ \frac{z^2}{r^2} \frac{\partial}{\partial \lambda} \int dx e^{-i\lambda x} J_0(x) +$$

$$+ \lambda \int dx e^{-i\lambda x} J_0(x) \} , \tag{46}$$

where $\lambda = ct/r$ ($>1$). By making use of Equation (34), we obtain

$$\psi(\boldsymbol{R}, t) \simeq \frac{1}{4c^2} \frac{(c^2 t^2 - r^2 - z^2) ct}{(c^2 t^2 - r^2)^{3/2}} \tag{47}$$

for $ct > r$ (i.e., $c^2t^2 > r^2$, and $c^2t^2 > r^2 + z^2$). This is precisely the result obtained previously [9]. In order to account for the small difference between the two velocities, from the denominator in Equation (42) we may infer that $K$ should be replaced with $Kv/c$ in the exponent of Equation (43), which amounts to replacing the time $t$ with the retarded time $\tau = tc/v$. The mainshock exhibited by Equation (47) is shown schematically in Figure 2.

We can check via direct calculations that

$$\ddot{\psi} - c^2 \Delta \psi = \ddot{\psi} - c^2 \left( \frac{\partial^2 \psi}{\partial r^2} + \frac{\partial \psi}{r \partial r} + \frac{\partial^2 \psi}{\partial z^2} \right) = 0 \ , \tag{48}$$

except for $z \to 0$, $r \to ct$. The singularity at $ct = r$ in Equation (47) (for $z = 0$) arises from the sharpness of the $\delta$-functions of the source term in Equation (29). It can be smoothed out by replacing $ct - r$ with $l$, where $l$ is an infinitesimal distance.

The solution given by Equation (47) has a spherical wavefront $r^2 + z^2 = c^2t^2$; it has a rapid variation with $r$ and a rather slow variation with $z$, so it may be viewed as a quasi-cylindrical wave, with a wall-like structure. It corresponds to the seismic mainshock. We can see that the potential given by Equation (47) and its spatial derivatives (the displacement) look like an abrupt wall with a long tail, propagating with velocity $c$ in the interior of the Earth and on its surface. Actually, we obtain two such walls, corresponding to the two primary $P$ and $S$ waves, propagating with velocities $c_{l,t}$.

## 7. Site Response and Inhomogeneities

The $\delta$-functions occurring in these problems should be viewed as highly peaked functions over a small region. For instance, $\delta(\boldsymbol{R})$, which occurs in the tensorial force acting in the earthquake focus (Equation (1)), is approximately $1/l^3$ over a small region with dimension $l$, where $l$ is of the order of the dimension of the seismic focus. The cutoff length $l$ also occurs in the primary seismic waves derived above (Equations (28)). It is related to the elastic energy stored in the seismic focus and released during an earthquake. A measure of the seismic energy is the earthquake (moment) magnitude $M_w$, such that, for instance, for an earthquake with magnitude $M_w = 7$, we obtain $l = 316$ m (for a density $\rho = 5$ g/cm$^3$ of the Earth and an average velocity $c = 5$ km/s of the elastic waves). However, the extension of the spot left by the seismic waves on the Earth's surface is much larger. This is so because of the energy loss suffered by the seismic waves during their propagation through the Earth.

The results presented above relate to a homogeneous and isotropic medium, while the Earth is recognized as inhomogeneous and anisotropic. The spatial distribution of a wave is characterized by its Fourier transform. Let us take a far-field seismic wave of the form

$$u = \frac{\delta'(R - ct)}{R} \ ; \tag{49}$$

its Fourier transform is

$$u(\boldsymbol{K}) = \int d\boldsymbol{R} \frac{\delta'(R - ct)}{R} e^{-i\boldsymbol{KR}} = -4\pi \cos cKt. \tag{50}$$

When encountering inhomogeneity, this wave sets in motion its particles, and even the inhomogeneity as a whole. Consequently, the incident wave loses energy, which is transferred to the inhomogeneity, which, in turn, generates secondary waves; part of the energy may be dissipated. Obviously, the effect is larger for small wavelengths, which are numerous, due to the large number of distinct directions for a large $K$. This is the wave scattering, with a possible energy loss (absorption). It is reasonable to assume that inhomogeneities are distributed relatively uniform, over their various size and mean separation distances. Consequently, we expect a secondary (scattered) diffuse radiation, with a large content of small wavelengths. It follows that the incident wave content is

diminished isotropically, with a larger weight for small wavelengths. This amounts to modifying the Fourier transform given above according to

$$u(\mathbf{K}) = -4\pi e^{-\alpha K} \cos cKt \ ,$$
(51)

where the parameter $\alpha$ characterizes the inhomogeneities' distribution. The resulting reverse Fourier transform is

$$u = -\frac{4\pi}{(2\pi)^3} \int d\mathbf{K} e^{-\alpha K} \cos cKt e^{i\mathbf{K}\mathbf{R}} =$$

$$= \frac{1}{\pi R}\frac{\partial}{\partial R}\frac{\alpha}{(R-ct)^2+\alpha^2} \ ,$$
(52)

where we retain only the retarded waves. We can see that we recover the incident wave given by Equation (49) in the limit $\alpha \to 0$. The effect of the inhomogeneities, included in the parameter $\alpha$, is a flattening of the $\delta'$-incident wave, which receives a larger width $l_0 = 2\alpha/\sqrt{3} > l$ and a smaller height $\simeq 1/l_0^2$. The ratio $l_0/l$ may attain values of the order of ten [16]. The scissor-like structure of the primary seismic waves shows that the frequency content of these waves is mainly centered on a single frequency, of the order of the wave velocity to the dimension of the seismic focus ($c/l_0$). Therefore, the Fourier analysis of the primary waves may give an estimate of the dimension of the focus [16].

The *P* and *S* seismic waves and the mainshock derived above are the seismic motion, generated by a point-like seismic focus acting for a short time interval. These results are obtained by assuming a homogeneous and isotropic elastic medium (the Earth). This assumption is valid for an average of the elastic properties of the medium. As long as we are interested in the overall, average behavior of the elastic motion, this is a satisfactory hypothesis. However, if we are interested in the local elastic motion, the elastic particularities of the site should be taken into account. In a simple model, any site may be viewed as a damped harmonic oscillator, with frequency $\omega_g$, connected by elastic forces to its surroundings. The seismic motion acts as an external force upon such an oscillator. The resulting motion is the ground motion. It consists of the original, scissor-like primary seismic waves and the wall-like seismic mainshock, over which the damped $\omega_g$-oscillations of the site are superposed; these damped oscillations are the seismic response of the site. In the ground motion, the long tail of the seismic mainshock is governed by the damping coefficient of the site. In turn, the ground motion acts as an external force upon the seismographs (or the structures built on the Earth's surface). As a simple model, we may take a linear damped harmonic oscillator for the seismograph, with frequency $\omega_s$, such that the seismograms record the $\omega_{g,s}$-oscillations, superposed over the original seismic motion. All these results are included in Ref. [16], where an estimation is also given of the maximum (peak) values of the ground motion displacement, velocity and acceleration, which may be useful as input parameters for seismic hazard studies.

## 8. Concluding Remarks

The primary seismic waves and the seismic mainshock are derived for a homogeneous and isotropic half-space with a plane surface (the Earth), by solving the elastic wave equation with the seismic tensorial force acting for a very short time interval (time-impulse) in a localized (point-like) seismic focus. The solution is the seismic motion. It is known as the Seismological, or Lamb's, Problem. The results reported previously are re-derived by using a new method, which emphasizes the regularization procedure. The time-impulse and point-like tensorial force is a combination of a temporal $\delta$-function and derivatives of a spatial $\delta$-function (Equation (1)). The equation of the elastic motion with such a source term may be called a singular equation. The singular nature of such a source leads to singular, improper solutions and unphysical contributions. Therefore, a regularization procedure is necessary. The regularization procedure employed here includes both a Coulomb-potential-type regularization, due to the use of the Fourier expansions, and a seismic wave regularization, specific to the singular elastic wave

equation. This regularization procedure amounts to using a long wavelength cutoff (Coulomb potential), an approximation via a continuous-function series for the stepwise $\theta$-function, the removal of superfluous solutions of the free-wave equation and a small cutoff time/length for the peaked temporal/spatial $\delta$-distribution.

In addition, a summary is provided for the effect of the inhomogeneities and the site response in determining the ground motion and the seismographs' recordings.

The primary waves (Equations (27) and (28)) and the mainshock (Equation (47)) presented in this paper correspond to a single, localized seismic focus. It may happen that the seismic energy is released by a succession of localized foci (ruptures), separated by short times and short distances in the focal region. This is a structured focus, discussed in detail in Ref. [10]. Since the equations are linear, the solution for a structured focus is a superposition of primary waves and mainshocks, like those given in this paper. A more complicated situation is a propagating focus, also discussed in Ref. [10].

**Funding:** This work was carried out within the Program Nucleu SOL4RISC, funded by the Romanian Ministry of Research, Innovation and Digitization, project no. PN23360201 and PN23360101.

**Data Availability Statement:** No data were used.

**Acknowledgments:** The author is indebted to the colleagues at the Institute of Earth's Physics, Magurele, and to the members of the Laboratory of Theoretical Physics, Magurele, for many enlightening discussions.

**Conflicts of Interest:** The author declares no conflict of interests.

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
