# Peer review of "Seismological Problem, Seismic Waves and the Seismic Mainshock"

_mathematics, doi:10.3390/math11173777_

Round 1

Reviewer 1 Report

The paper studied the elastic wave equation with the seismic tensorial force in a homogeneous and isotropic medium, and provides a detailed procedure of the regularization. In addition, this article also discusses the generation of secondary waves, mainshock and a summary information for the effect of the inhomogeneities and the site response. Although the topic is interesting and meaningful, this paper has a number of drawbacks, which have to be addressed.

1) The authors proposed a different method of solving the Navier-Cauchy equation, which throws more light upon the singular, unphysical behaviour of the solution. However, the comparison and advantages between the new method proposed in this paper and previous literature have not been given. Including the secondary waves studied later, they have been studied by many people. So what are your new results? It should be emphasized.

2) All symbols used in the paper should be explained when first used. Many symbols (θ, μ, δ, l and others) in the article were not explained when first used. In addition, the meaning of the same symbol should be consistent. For example, the symbol l does not give any meaning when first used in Section 3, and represents an infinitesimal distance in Section 6, and represents the order of the dimension of the seismic focus in Section 7.

3) If certain research contents have already been published, it does not need to appear in this article or should be briefly mentioned. For example, do the contents of Sections 3 and 7 seem to have a source? The differences in the research content of this article should be emphasized.

4) There is only one figure in the entire paper, which is the schematic representation of a typical seismogram. Many formulas are derived in this paper, but no results are shown in the form of images. I suggest adding examples or actual seismic data to better showcase the research results.

5) Except for the author's own references, all other references have been published for decades or even more than a hundred years. The research on seismic engineering and seismic waves has always been a focus of attention and exploration by scholars, and there is no lack of reference literature. Suggest adding references closely related to your research, especially in recent years.

Overall, the paper proposes an interesting and approximate analytical solution for a complex seismological problem. At the same time, there are some problems with this paper. This paper will not be accepted unless it is thoroughly revised.

 Moderate editing of English language required

Reviewer 2 Report

The paper describes a rigorous mathematical formulation for the generation of primary and secondary seismic waves from the basic elastic wave propagation model. The formulation is also extended to evaluate the ground motion characteristics for a site-specific situation. The manuscript is well-written and is of interest to the seismology community. However, the presentation needs to be improved. Several equations are presented without context which makes it hard to follow. Please provide sufficient background to these mathematical formulations. I also suggest adding more diagrams, figures, schematics, etc. to improve the quality of the presentation. 

1. The novelty of the research is unclear. Please provide a more detailed literature review in the introduction section to outline how the current research sheds light on a gap in the existing literature. 

2. Lines 46-48: Could you explain the 'wall' and 'tail' with the help of a diagram for better visualization?

3. Equation 2: How is the displacement vector ui different from uj? Please add in text.

4.  Many equations don't describe what the variables or symbols mean. Please correct. For example, ki, kj, in equation 3. It would be great to mention these even if they are well-known within the community for readers from other scientific communities. 

5. A large number of equations are presented in the manuscript. It is hard to follow the transitions between different equations especially when many terms aren't described (For example, in line 72 and equation 6, the meaning of ε is not described). There are several similar occurrences. This is a major shortcoming of the current version. I would suggest adding more figures, flowcharts, etc. to make understanding of the equations more clear. 

6. Line 75:  Not clear what is meant by "We need to give a meaning to this integral". Is the end goal to convert this to a definite integral that can be computed?

7. Major revisions are needed in sections 3 and 4 describing coulomb-potential regularization and seismic-wave regularization. These techniques are not clearly described. Recommend adding a brief introduction to the technique in the text first before delving into the equations. 

8. Section 5 : please add a schematic figure showing the propagation of the primary waves, production of secondary waves. In the figure, denote the position vector (r) and direction of z and where z=0 lies. 

9. Section 7: When converting the seismic motion to a site-specific ground motion, what is the effect of the frequency content of the incident seismic motion on the site? 

10. Please justify the use of a single parameter α (equation 51) to model the inhomogeneity distribution between the source and the receiver (seismogram). Can α be computed in a real-world scenario if the specific site conditions are known?

11. Can the mathematical formulations presented in the manuscript be used when several consecutive ruptures are present separated by both distance and time? If not, what changes would be required? Please include some comments in the conclusion section. 

Reviewer 3 Report

The manuscript entitled "Seismological problem, seismic waves and the seismic mainshock" is based on analytical research. 

In this paper, the author derived the primary seismic waves and seismic mainshock for homogeneous and isotropic half-space. In this paper, the Coulomb-potential regularization procedure as well as the seismic-wave regularization procedure were used to remove unphysical contributions arising from the singular nature of the source term in the governing equation.

The paper is well organized and well written. The author described the solution process in sufficient detail, although some parameters used in the equations are not defined. Also, in section "2", the main difference between the present article and the author's previous research and the importance of addressing the problem have been pointed out (It is better to discuss it in more detail in the introduction). However, up-to-date sources are not used in the introduction.

Round 2

Reviewer 1 Report

The authors responded well to the reviewers' comments. I still believe that a concrete and detailed example can be used to describe the topic more vividly, especially a specific result or image of the equation. However, this is just a suggestion for future research. Overall, the article is acceptable.